# Analysis of the Factors Affecting Germination of *Cnidium monnieri* Seeds and Its Endogenous Inhibitory Substances

**DOI:** 10.3390/plants14243801

**Published:** 2025-12-13

**Authors:** Qiantong Wei, Shulian Shan, Sirui Zhao, Chongyi Liu, Feng Ge, Hongying Cui, Fajun Chen

**Affiliations:** 1State Key Laboratory of Agricultural and Forestry Biosecurity, College of Plant Protection, Nanjing Agricultural University, Nanjing 210095, China; 2023802144@stu.njau.edu.cn (Q.W.); 2024802146@stu.njau.edu.cn (S.S.); 20220303002@stu.sxau.edu.cn (S.Z.); liuchongyi@stu.njau.edu.cn (C.L.); 2Shandong Key Laboratory for Green Prevention and Control of Agricultural Pests, Institute of Plant Protection, Shandong Academy of Agricultural Sciences, Jinan 250100, China; gef@ioz.ac.cn

**Keywords:** *Cnidium monnieri*, seed germination, viability, absorbency, inhibitory substances, ecological application

## Abstract

*Cnidium monnieri* is a valuable functional plant with significant potential for green pest control. However, its large-scale application is limited by its low and uneven seed germination in fields. To determine the factors that affect the germination of *C. monnieri* seeds, we examined its seed viability, germination percentage and germination speed index (GSI) after seed-coat treatments, water permeability, and the types and activity of endogenous inhibitory substances in *C. monnieri* seeds. The results indicated that the seed viability of *C. monnieri* is 95%, but the germination percentage was relatively low (12.60%). Seed coat removal significantly enhanced both the germination percentage and the GSI, but had no significant effect on water absorption rate. Moreover, ethyl acetate extracts completely inhibited the seed germination of the control non-dormant *Brassica rapa* subsp. rapa, while diethyl ether extracts showed moderate suppression, and petroleum ether extracts exhibited the weakest effect. And the three endogenous inhibitory substances, i.e., dibutyl phthalate, 2,6-di-tert-butylphenol, and 2,4-di-tert-butylphenol significantly reduced the seed germination, seedling height and root length of *B. rapa*, indicating their high inhibitory efficiency on seed germination. Our study demonstrates that the mechanical barrier of the seed coat and the presence of potent endogenous germination inhibitory substances are the key factors influencing the germination of *C. monnieri* seeds. These findings provide a theoretical basis for promoting seed germination of *C. monnieri*, which enhance its application value as functional plant for green pest control.

## 1. Introduction

*Cnidium monnieri* is a plant species of the genus Cnidium within the family Apiaceae, primarily distributed across the East, North and Northeast China [1]. It has been demonstrated that *C. monnieri* has significant ecological functions and application values. Firstly, as a traditional Chinese medicinal resource, the fruit of *C. monnieri* contains osthole, a significant coumarin compound exhibiting anti-inflammatory and antibacterial pharmacological activities in medical fields [2]. In agriculture, it shows notable insecticidal and bacteriostatic effects, indicating potential for novel biopesticide development [3]. Secondly, from a landscape ecology perspective, *C. monnieri*, as an important functional plant, possesses unique ornamental values and application advantages [4]. This plant exhibits remarkable ecological adaptability and contributes to the stability of flower-visiting insect communities [5]. Field trials indicated that planting *C. monnieri* substantially increased the populations of natural enemies (e.g., *Harmonia axyridis* and *Propylea japonica*) in wheat and maize fields, thereby significantly reduced the aphid abundances and enabled ecological pest control in the field [6]. In agricultural applications, planting *C. monnieri* can effectively reduce the use of chemical pesticides while improving crop quality, thus achieving dual improvements in production and ecological benefits [7]. Consequently, field cultivation of *C. monnieri* can substantially enhance farmland ecosystem service functions, aligning with contemporary trends towards diversified agricultural development. However, the industrial development of *C. monnieri* is severely constrained by its seed biology. Previous observations revealed that *C. monnieri* seeds exhibited typical dormancy characteristics, with germination percentage often <20%, which extended the emergence period (15–20 days at low temperatures, and 5–10 days under suitable temperatures) and resulted in uneven seedling emergence, severely restricting the seed supply security in the planting base and practical application of *C. monnieri* in large-scale fields [5].

Seed dormancy is one of the characteristics of plant adaptation to its ecological environment, possessing significant ecological values [8]. Seed dormancy, as part of a modern classification system, has undergone substantial re-evaluation over the past two decades. According to the updated framework proposed by Baskin (2014) [9], Baskin and Baskin [10], dormancy is categorized into morphological dormancy (MD), physiological dormancy (PD), morphophysiological dormancy (MPD), physical dormancy (PY), and combinational dormancy (PY + PD). Numerous studies have shown that species in the Apiaceae family typically exhibit MPD, characterized by underdeveloped embryos during dispersal together with a physiological component that prevents germination (e.g., *Cyclospermum leptophyllum* [11], *Angelica keiskei* [12], and *Ferula gummosa* [13]). *C. monnieri* seeds have also been explicitly classified as having MPD, requiring both embryo growth and the release of physiological dormancy for successful germination [14]. Previous research further demonstrated that cold stratification promoted embryo elongation and significantly increased germination percentage in this species, confirming the simultaneous presence of MD and temperature-responsive PD [15].

Given that the MPD seeds usually exhibit low germination despite being physiologically viable, it is essential to distinguish true dormancy from seed mortality when evaluating the seed germination of *C. monnieri* [16]. Seed viability assessment, such as the tetrazolium chloride (TTC) staining test, provides a reliable method for determining whether non-germinating seeds are alive [17]. TTC analysis is therefore an important prerequisite for interpreting dormancy-related germination responses, particularly in species with complex dormancy mechanisms such as those in the Apiaceae family [18]. In the context of *C. monnieri*, assessing viability is crucial to confirm that low seed germination reflects just MPD rather than reduced seed vigor.

After the radicle of MPD seeds has fully developed, the mechanical resistance from the embryo-covering layers continues to restrict the embryonic growth potential, resulting in insufficient growth force to overcome the constraint of peripheral tissues. Therefore, the seed coat is a crucial factor influencing seed MPD [16,19,20]. Embryos must overcome inhibition of the seed itself to break MPD and germinate. The seed-covering layer(s) mainly involved the dead outer testa or pericarp and the living inner testa, endosperm or the radicle [16]. For example, the removal of seed coat in *Pastinaca sativa* seeds has been observed to accelerate embryonic growth compared to intact seeds, which indicates that seed coat acts as a germination constraint, together with embryonic growth before germination, determines the timing of seed germination [21]. Similarly, *Angelica palustris* seeds showed improved germination following scarification [9]. However, how seeds coat affects the MPD of *C. monnieri* seeds remains unclear.

In addition, many seeds contain endogenous germination inhibitory substances–secondary metabolites or volatiles that suppress cell elongation and delay radicle emergence [22,23]. These compounds can be extracted into different organic phases according to their solubility and subsequently tested using non-dormant indicator species such as *Brassica rapa subsp*. rapa [24]. Aqueous extracts from *Pimpinella anisum* and *Trachyspermum ammi* seeds significantly reduced the germination percentage of *Lactuca sativa* [25]. Previous studies found that dibutyl phthalate, 2,6-di-tert-butylphenol and 2,4-di-tert-butylphenol inhibited seed germination percentage through allelopathic effects, thereby affecting MPD. For example, 2,6-di-tert-butylphenol and dibutyl phthalate interfered with the normal growth of *Alternanthera philoxeroides* and *B. rapa* by inducing abiotic stress, which was manifested as reduced plant height, thinner stems, diminished root activity, and ultimately impaired physiological function [26,27]. 2,4-di-tert-butylphenol significantly inhibited the seed germination of *Atractylodes macrocephala* [28]. Although scattered reports concerning *C. monnieri* germination behavior, the relative contribution of inhibitory compounds remains insufficiently understood within the MPD conceptual framework.

Given the potential involvement of seed-coat structure and endogenous inhibitory substances in regulating the germination process of most plant seeds, this study aimed to investigate the roles of seed coat modification and inhibitory substance on the seed germination of *C. monnieri*. By integrating germination assays with gas chromatography-mass spectrometry (GC-MS)-based chemical profiling, we seek to evaluate whether there are endogenous inhibitory substances to affect the seed germination of *C. monnieri*. Clarifying these regulatory mechanisms will improve the understanding of germination barriers in *C. monnieri* and contribute to the development of effective propagation strategies for large-scale cultivation.

## 2. Materials and Methods

### 2.1. Plant Materials

The seeds of *C. monnieri* were collected from the field station of Shandong Academy of Agricultural Sciences located in Jinan, Shandong Province of China (36.97° N, 116.99° E). Pure seeds were obtained by sieving to remove impurities and damaged seeds, ensuring the uniformity of the seeds, which were then air-dried at room temperature and stored for following experiments. In general, *B. rapa* seeds are frequently used in seed dormancy inhibitor validation experiments because of their non-dormant nature, high germination percentage, easy germination, and short germination period [29]. In this study, the seeds of *B. rapa* (cv. Zaofeng No. 1) were commercially bought from the local market for the following experiment to validate the inhibitory effects of the organic solvent extracts from *C. monnieri* seeds on seed germination. For *B. rapa*, its germination percentage was ≥95.0%, moisture content was ≤7.0%, and purity was ≥98.0%.

### 2.2. Experimental Methods

#### 2.2.1. Viability Test of *C. monnieri* Seeds

Seed viability of *C. monnieri* was tested by using the 2,3,5-triphenyltetrazolium chloride (TTC; Beijing Solarbio Science & Technology Co., Ltd., Beijing, China) method [30]. Fifty seeds of *C. monnieri* were randomly selected and placed in a 9 cm diameter Petri dish. These seeds were fully soaked in 50 mL of distilled water and incubated at a constant temperature of 25 °C for 24 h in darkness to achieve saturation, and then the seed coats were longitudinally incised along the seed ridge to expose embryonic tissues, in order to enhance TTC penetration. The seeds were then treated with 0.3% TTC staining solution, which was allowed to remain submerged at 25 °C in a dark environment for 6 h. After staining, the solution was drained and the treated seeds were thoroughly rinsed with tap water before microscopic examination under a stereomicroscope (Nikon SMZ745T stereo microscope; Nikon Corporation, Tokyo, Japan) to assess embryo coloration. Seeds exhibiting red staining were identified as viable with four replicate trials and 50 seeds per trial.

#### 2.2.2. Germination Assay of Differentially Treated Seed Coats of *C. monnieri*

After 12 h of soaking, the seeds of *C. monnieri* were sterilized and then subjected to different seed-coat treatments under sterile conditions according to the experimental design with three seed-coat treatments, including intact seeds (CK), cracked seeds (i.e., seeds with cracked seed coats), and decorticated seeds (i.e., seeds with entire removed coats). Simultaneously, all seeds were positioned with the radicle oriented downward on Petri dishes lined with three layers of filter paper. All the experiments were conducted in a controlled-environment growth chamber (Model RXZ-500; Ningbo Jiangnan Co., Ltd., Ningbo, China) under conditions of 25 ± 1 °C temperature, 14L/10D, and 60 ± 5% RH. Each treatment included 100 seeds of *C. monnieri* with three replicates. Germination percentage and growth status of *C. monnieri* seeds were regularly monitored every day until the germination percentage was recorded after 15 days. And the seeds were considered germinated when the radicle length exceeded the embryo’s length. The germination speed index (GSI) was also calculated simultaneously with the germination test.

Germination percentage: G = (N/A) × 100%, where G is the germination percentage, N is the number of germinated seeds, and A is the number of seeds in the sample [31].

Germination speed index (GSI) is calculated by daily counting of germinated seeds and using the formula GSI = ΣPi/Di, where GSI is the germination speed index, Pi is the number of seeds germinated on the i-th day, and Di is the number of days from the start of the test to the i-th day [32].

#### 2.2.3. Determination of Water Permeability of Differentially Treated *C. monnieri* Seeds

The seeds of *C. monnieri* were weighed by using a analytical balance (Model: SQP; Sartorius AG, Göttingen, Germany; Range: 220 g; Precision accuracy: 1 mg). Fifty seeds were randomly selected and weighed separately from each of the seed-coat treatments with intact, cracked, and decorticated seeds, and then soaked in a constant-temperature water bath at 25 °C for water absorption. The treated seeds were weighed every hour for the first 6 h, and every 12 h from 6 to 30 h, and then every 24 h until reaching constant weight at 102 h. The water absorption rate was calculated with three replicates per treatment and 100 seeds per replicate, which was calculated using the following formula that W = (B − G)/G × 100%, where W represented the seed water absorption rate (%), B denoted the weight of the seed after water absorption (g), and G indicated the initial weight of the seed (g) [33].

#### 2.2.4. Determination of Inhibitory Effects of Different Organic Phase Extracts from Different Seed Parts of *C. monnieri* on the Viability of Non-Dormant *B. rapa* Seeds

To verify the presence of inhibitory substances in *C. monnieri* seeds and to identify their distribution within the seed, 10 g seeds of intact seeds, seed coats, and seed kernels were ground and soaked in 80% methanol solution, respectively, and stored airtight at 4 °C and stirred periodically to ensure thorough mixing. The extracts were vacuum-filtered every 24 h, and the solid residues were re-extracted with fresh 80% methanol solution. This extraction process was repeated three times, and the filtrates were combined to obtain the methanol extracts from the intact seeds, seed coats, and seed kernels, respectively. These methanol extracts were subsequently subjected to a systematic solvent partitioning procedure. This separation produced three different phases from each extract, including petroleum ether phase, ether phase, and ethyl acetate phase.

For each organic phases, 3 mL extract was placed in a Petri dish with two layers of filter paper. Distilled water was replenished at regular intervals over 48 h to ensure seed saturation. The respective control phase was established using equivalent volumes of distilled water and their corresponding pure organic solvents. For this experiment, the non-dormant *B. rapa* seeds were used as control to assess the inhibitory effects of different organic phase extracts from different seed parts of *C. monnieri* on seed viability of *B. rapa*. Each treatment was replicated three times with 30 seeds of *B. rapa* per replicate placed in the Petri dishes, which were pre-soaked in distilled water for 2 h. The germination percentage of *B. rapa* seeds was measured after 72 h, and those seeds were considered germinated when the radicle length exceeded half the length of the seed.

#### 2.2.5. Extraction, Separation and Identification of Endogenous Inhibitory Substances from *C. monnieri* Seeds

The extract from each organic solvent phase was centrifuged at 4000 r·min^−1^ for 5 min, the supernatant was collected and vacuum-filtered, and then concentrated to 50 mL under vacuum using a rotary evaporator, which produced the concentrated organic phase. The conical flask was then sealed with plastic film and stored at 4 °C for the following treatments. Subsequently, the concentrated organic phases were subjected to vacuum concentration via rotary evaporation, and the resulting solids were dissolved in the corresponding organic solvent to a final volume of 1.5 mL. And the specific components of the three organic phase extracts (petroleum ether, diethyl ether, and ethyl acetate) were analyzed and identified by gas chromatography–mass spectrometry (GC-MS). The analysis was performed on an Agilent 7890B GC-MS system (Agilent Technologies, Santa Clara, CA, USA) equipped with a WAX quartz capillary column (100 m × 250 μm × 0.2 μm; Agilent J&W DB-Wax). The oven temperature was programmed from 0 to 225 °C at 10 °C·min^−1^. Helium was used as the carrier gas, and the injection port temperature was set to 250 °C. The mass spectrometer was operated in EI mode at 70 eV, with ion source temperature of 230 °C and quadrupole temperature of 150 °C, scanning from 50–500 amu with 1 μL injection volume. Mass spectra of the compounds were compared with the NIST MS database and co-injected with authentic standards. The identification results were supplemented by manual interpretation. And relative abundances were calculated using the peak area normalization method.

#### 2.2.6. Impact of Three Key Inhibitory Substances on the Germination and Growth of *B. rapa* Seeds

According to the GC-MS appraisal results (i.e., 3.5), we identified the three metabolites, e.g., 2,6-di-tert-butylphenol, dibutyl phthalate, and 2,4-di-tert-butylphenol. Subsequently, these three commercial standards (Cas No. 128-39-2/84-74-2/96-76-4) were procured from Shanghai Macklin Biochemical Technology Co., Ltd. (Shanghai, China). The 2,6-di-tert-butylphenol, dibutyl phthalate, and 2,4-di-tert-butylphenol were diluted in ethanol solutions to concentrations of 1, 10 and 100 mg/L, respectively. Ethanol solution was used as a blank control for the germination experiment of *B. rapa* seeds.

### 2.3. Statistical Analysis

All statistical analyses were conducted using SPSS 13.0 (SPSS Inc., Chicago, IL, USA). Two-way factorial ANOVAs were used to validate the inhibitory effects of different organic phase extracts (those from the phases of petroleum ether, diethyl ether, and ethyl acetate) from different seed parts (including intact seeds, seed coats, and seed kernels) of *C. monnieri* on the germination percentage of the non-dormant *B. rapa* seeds in the germination inhibition bioassay. One-way ANOVA was used to evaluate the effects of seed coat treatments (intact, cracked, and decorticated) on the germination percentage, germination speed index, and water absorption of *C. monnieri* seeds, with respect to the relative abundance of different compounds within the same organic phase extract, as well as to compare the effects of different germination inhibitory substances on the germination percentage, seedling height, and root length of *B. rapa* seeds. And the post hoc multiple comparisons were performed using Tukey’s test at *p* < 0.05.

## 3. Results

### 3.1. Seed Viability of C. monnieri

The seeds of *C. monnieri* are narrow-elliptic or elliptic, slightly flat, with a somewhat rough and brown surface, and the ventral surface is flat and concave (Figure 1). After the TTC staining treatment, those non-viable seeds remained unstained (Figure 1A), whereas those viable seeds were stained red (Figure 1B). And the viability of seeds with stained red color was high as 95% of the total seeds treated with the TTC staining.

### 3.2. Effects of Different Seed-Coat Treatments on the Germination Percentage of C. monnieri Seeds

In the experiment, the *C. monnieri* seeds subjected to various seed-coat treatments (i.e., intact, cracked and decorticated seeds) began to germinate on blank culture medium within 3–5 days. The seed-coat treatment significantly affected the germination of *C. monnieri* seeds (F = 7.86, *p* < 0.01; Table 1). And the germination percentage of cracked seeds (21.00%) and decorticated seeds (29.20%) increased by 66.67% (*p* > 0.05) and 131.74% (*p* < 0.05) compared to that of intact seeds (12.60%), respectively (Figure 2A).

The seed-coat treatment significantly affected the germination speed index (GSI) of *C. monnieri* seeds (F = 7.25, *p* < 0.01; Table 1). The GSI value of decorticated seeds (38.46) increased by 141.88% compared to that of intact seeds (15.90) (*p* < 0.05; Figure 2B). And the GSI value of cracked seeds (22.06) increased by 38.74% compared to the intact seeds (*p* > 0.05; Figure 2B).

### 3.3. Effects of Different Seed-Coat Treatments on the Water Absorption Rate of C. monnieri Seeds

The water absorption trends of *C. monnieri* seeds were generally consistent for the three seed-coat treatments of intact, cracked, and decorticated seeds, which were divided into three stages, i.e., rapid (0–6 h), slow (6–30 h), and stable (30–102 h) absorption stages (seen in Figure 3). During the initial rapid absorption stage, the intact, cracked and decorticated seeds displayed rapid water uptake with absorption rates reaching 96.62%, 62.63% and 33.42%, respectively. In the slow absorption stage, the water absorption rates of the intact, cracked and decorticated seeds initially increased gently and then declined with seed germination. During the subsequent stable absorption stage, the water absorption rates of all three types of seeds gradually approached uptaking saturation, with final absorption rates of 60.92%, 55.27%, and 52.63% for the intact, cracked, and decorticated seeds, respectively (Figure 3). This indicates that the intact seeds of *C. monnieri* have good water absorption and persistence capacity.

Seed-coat treatment significantly affected the water absorption rates of *C. monnieri* seeds at 2 h (F = 22.85, *p* < 0.01), 3 h (F = 17.41, *p* < 0.01), 4 h (F = 6.44, *p* < 0.05), 5 h (F = 9.64, *p* < 0.05), 6 h (F = 6.91, *p* < 0.05), 18 h (F = 9.63, *p* < 0.05) and 54 h (F = 14.064, *p* < 0.001) of water absorption (Table 1). There was no significant difference in the water absorption rate between the intact and cracked seeds (*p* > 0.05), whereas the water absorption rate of decorticated seeds was significantly lower than that of intact seeds, with a reduction of 47.75–71.70%, 76.97%, and 36.65% at 2–18 h, 54 h, and 102 h, respectively (*p* < 0.05; Figure 3).

### 3.4. Inhibitory Effects of Different Organic Phase Extracts from Different Parts of C. monnieri Seeds on the Germination of Non-Dormant B. rapa Seeds

Different seed parts of *C. monnieri* (F = 47.398, *p* < 0.001) and different organic phase extract (F = 548.084, *p* < 0.001), and their interactions (F = 42.278, *p* < 0.001) all significantly affected the germination percentage of non-dormant *B. rapa* seeds (Table 2). As shown in Figure 4, compared to the control group, the ethyl acetate phase extracts from both whole seeds and seed coats completely inhibited the germination of *B. rapa* seeds (germination percentage = 0%). The diethyl ether phase extract from seed coats and the diethyl ether phase extract from kernels both significantly inhibited *B. rapa* seed germination (1.11%; *p* < 0.05, Figure 4). Furthermore, the inhibitory effects of the petroleum ether phase extracts from whole seeds, seed coats, and kernels, as well as the ethyl acetate phase extract from kernels showed no significant difference compared to that of the control (*p* > 0.05; Figure 4).

### 3.5. Identification of Key Germination-Inhibitory Components in C. monnieri Seeds

The GC-MS identification chromatograms on the extract components of the three phases of diethyl ether, ethyl acetate, and petroleum ether from *C. monnieri* seeds were shown in Figure 5. By comparing with the GC-MS standard mass spectral library, a total of 19 compounds belonging to 8 major classes were identified, including alkanes, alkenes, alcohols, oleic acids, coumarins, esters, phenols, and aromatic hydrocarbons (Table 3). Moreover, there was markedly different in the relative percent among the extracted substances from the diethyl ether phase (F = 2.47, *p* = 0.092) and the ethyl acetate phase (F = 2.70, *p* = 0.069), while the relative percentage of extraction substances of petroleum ether fraction was not significant (F = 1.12, *p* = 0.39; Table 3).

#### 3.5.1. Identification of the Extract Components of Diethyl Ether Phase

GC-MS analysis identified six main compounds in the diethyl ether phase extracted from *C. monnieri* seeds, i.e., 2,6-ditert-butyl--phenol (39.78%), dibutyl phthalate (22.08%), p-xylene (13.76%), cyclooctatetraene (3.63%), methyl 14-(2-octylcyclopropyl) tetradecanoate (2.20%), and cis-9-octadecenoic acid (1.10%), and the relative percent of 2,6-ditert-butyl-phenol was significantly higher than that of other five components (*p* < 0.05; Table 3).

#### 3.5.2. Identification of the Extract Components of Ethyl Acetate Phase

GC-MS analysis also identified six main compounds in the ethyl acetate phase extracted from *C. monnieri* seeds, i.e., dibutyl phthalate (47.03%), osthole (11.60%), 9,9′-[1,4-phenylenedi-2,1-ethanediyl] bis [2,3,6,7-tetrahydro-1H,5H-benzopyrano (3)] etradicine (8.27%), 2,4-di-tert-butylphenol (7.44%), 1,2-15,16-diepoxyhexadecane (3.54%), and 1,3-diacetoxy-2-propanyl laurate (2.18%), and the relative percent of dibutyl phthalate was significantly higher than that of other five components (*p* < 0.05; Table 3).

#### 3.5.3. Identification of the Extract Components of Petroleum Ether Phase

GC-MS analysis identified eleven main compounds in the petroleum ether phase extracted from *C. monnieri* seeds, i.e., 1,2-15,16-diepoxyhexadecane (19.73%), osthole (19.08%), perilla alcohol (18.51%), cis-p-mentha-1(7),8-dien-2-ol (10.50%), 2,6-di-tert-butyl--phenol (10.43%), methyl oleate (9.06%), isobornyl acetate (4.20%), 9,9′-[1,4-phenylenedi-2,1-ethanediyl] bis [2,3,6,7-tetrahydro-1H,5H-benzopyrano (3)] etradicine (2.44%), methoxyacetic acid,2-tridecyl ester (2.28%), 2,6,10-trimethyltetradecane (2.25%), and 6-methylhexadecane (1.53%), while there was no significant difference in the relative percentage of the 11 main compounds (*p* > 0.05; Table 3).

### 3.6. Dibutyl Phthalate, 2,6-Di-tert-butylphenol, and 2,4-Di-tert-butylphenol Exhibited Significant Inhibitory Effects on Seed Germination and Seedling Growth of B. rapa

2,6-di-tert-butylphenol and dibutyl phthalate were the most abundant compounds in *C. monnieri* seeds based on Section 3.5, while 2,4-di-tert-butyl-p-cresol is commonly identified as endogenous inhibitory substances. So these three inhibitory substances were selected to assay their inhibitory function on seed germination. Dibutyl phthalate (F = 83.27, *p* < 0.001), 2,6-di-tert-butylphenol (F = 5.19, *p* < 0.05), and 2,4-di-tert-butylphenol (F = 8.77, *p* < 0.01) significantly inhibited the seed germination percentage of *B. rapa* (Figure 6A). Compared to CK, at a concentration of 100 mg/L, these inhibitory substances significantly reduced the germination percentage by 25.56%, 8.89%, and 15.56%, respectively (*p* < 0.05; Figure 6A); and at a concentration of 10 mg/L, 2,4-di-tert-butylphenol also significantly inhibited the seed germination percentage of *B. rapa* by 66.00% (*p* < 0.05; Figure 6A).

Dibutyl phthalate (F = 105.79, *p* < 0.001), 2,6-di-tert-butylphenol (F = 145.72, *p* < 0.001), and 2,4-di-tert-butylphenol (F = 141.45, *p* < 0.001) also significantly affected the seedling height, with the inhibitory effect intensifying as the treatment concentration increased. Compared to CK, the seedling height significantly decreased by 3.42–4.49 mm, 2.68–3.91 mm, and 3.38–4.31 mm under 1–100 mg/L concentration of these three inhibitory substances, respectively (*p* < 0.05; Figure 6B). Similarly, the root growth of *B. rapa* was also significantly suppressed by dibutyl phthalate (F = 142.82, *p* < 0.001), 2,6-di-tert-butylphenol (F = 112.98, *p* < 0.001), and 2,4-di-tert-butylphenol (F = 132.08, *p* < 0.001), showing significantly reduction of 8.93–11.74 mm, 8.71–10.38 mm, and 8.73–10.27 mm compared to the CK, respectively (*p* < 0.05; Figure 6C).

## 4. Discussion

Seed dormancy is an important survival strategy of some plants during their long-term evolution, referring to the phenomenon where viable seeds fail to germinate under suitable conditions over a certain period [9]. Seeds of *C. monnieri* have been explicitly classified as having morphophysiological dormancy (MPD), in which germination is restricted by both an underdeveloped embryo and a physiological dormancy component [14]. In this study, *C. monnieri* seeds exhibited high viability (>95% based on TTC staining), but the germination percentage was only 12.60%, indicating that the low germination percentage is primarily due to dormancy rather than seed mortality or reduced vigor.

According to the MPD concept, seed coat-imposed constraints (including mechanical resistance and permeability restrictions) are constitutive factors of MPD dormancy. In plant species with MPD, the seed coat acts as an important germination constraint by restricting the permissive temperature window for embryo growth and radicle emergence [34,35]. For instance, *Heracleum sphondylium* exhibited increased germination after seed-coat weakening or partial removal, which enhanced radicle protrusion and oxygen availability without altering dormancy type [9]. In *Fritillaria taipaiensis*, the seed coat cells were tightly arranged and adhered closely to the endosperm. Before radicle emergence, the embryo must overcome the mechanical constraints imposed by both the endosperm and seed coat [36]. In this study, we found that the values of germination percentage (GP; 21.00%) and germination speed index (GSI; 22.06) of cracked seeds were higher than those of intact seeds (GP: 12.60%; GSI: 15.90), and were further increased in decorticated seeds (GP: 29.20%; GSI: 38.46). These findings indicate that the seed coat of *C. monnieri* acts as a mechanical constraint that impedes the protrusion of the morphologically mature radicle, thereby preventing germination. Previous studies have found that damaged seeds release substances such as reactive oxygen species (ROS) and nitric oxide (NO) as the emergency response, enabling dormant seeds to begin growing into seedlings before their stored reserves are consumed by saprophytic microorganisms [37]. Our study found that there was no significant difference in germination percentage between cracked seeds and intact seeds, possibly because the levels of ROS and NO produced by the seed coat cracking treatment did not reach the threshold required for germination. Furthermore, cracked seeds imbibed water more slowly than intact seeds, demonstrating that enhanced germination cannot be attributed to accelerated water absorption. Similar “slow-imbibition but high-germination” patterns have been reported in MPD species such as *Daucus carota* and *Foeniculum vulgare*, where water rapidly hydrates the pericarp but is not immediately available to the embryo [16,38]. The water absorption rate of cracked seeds of *Cryptotaenia japonica* showed no significant difference compared to that of intact seeds [39], which is consistent with our findings. All these findings indicate that reduced mechanical resistance in seed coat microenvironment play more decisive roles on the seed germination of *C. monnieri*.

Endogenous germination inhibitory substances can also enhance the physiological components of MPD in plants. Research had found that seed dormancy in *Heracleum moellendorffii* and *Epimedium wushanense* were caused by endogenous inhibitory substances [40,41]. Seed dormancy is determined by the balance between endogenous promoters and inhibitory substances, and breaking dormancy essentially involves the shift in this dynamic equilibrium toward germination-promoting factors [42]. The presence of germination inhibitory substances can be verified by applying diluted extracts containing these substances to non-dormant seeds, such as *B. rapa* seeds [43]. Endogenous inhibitory substances in seeds are partitioned into different organic phases based on their respective solubilities. The bioassays using petroleum ether, diethyl ether, and ethyl acetate phase extracts from *Onobrychis viciifolia* seeds on *B. rapa* seeds demonstrated that the ethyl acetate phase exhibited the highest inhibitory effect, which is consistent with our findings [23]. Zou et al. (2021) found that different extracts of *Paeonia ostii* seeds exhibited varying effects on the germination of *B.rapa* seeds, with the ether and ethyl acetate phases from the seed coats, along with the ether and methanol phases from the embryo [44]. Correspondingly, our results also showed that the diethyl ether and ethyl acetate phase extracts from both the seed coats and kernels of *C. monnieri* significantly inhibited *B. rapa* seed germination, whereas the petroleum ether phase extract exhibited a relatively weaker inhibitory effect on *B. rapa* seed germination. This aligns with the findings of Tang et al. (2024) regarding the differential inhibition of seed germination by various organic phase extracts containing endogenous inhibitory substances in *Symplocos paniculata* seeds [24]. It is apparent that different organic phases treatments have a significant inhibitory effect on the germination of *B. rapa*, which may be due to the type and/or concentration of germination inhibitory substances that vary at different stages. Moreover, our findings showed that the inhibitory effects of the organic phase extracts derived from whole seeds was lower than that from seed kernels and coats, but there was no significant difference between the organic phase extracts of seed kernels and coats. This indicates that the inhibitory substances in *C. monnieri* seeds are distributed in both the coats and the kernels.

Many substances, including ethylene, aromatic oils, alkaloids, and various organic acids, can significantly impede cell division, differentiation, and elongation, thereby delaying or suppressing seed germination [45,46]. Abscisic acid, 2,4-di-tert-butyl-p-cresol, dibutyl phthalate, and palmitic acid are commonly identified as endogenous inhibitory substances [45]. In our study, we also detected the three inhibitory substances, and 2,6-di-tert-butylphenol and dibutyl phthalate were the most abundant compounds across the various organic phase extracts from *C. monnieri* seeds. Yu et al. (2022) identified 2,6-di-tert-butylphenol and 2,4-di-tert-butylphenol as significant endogenous inhibitory substances, causing seed dormancy in *Agropyron mongolicum* [47]. And high concentrations of 2,6-di-tert-butylphenol and its isomer exerted inhibitory effects on seed germination of vegetable crops such as *Capsicum annuum*, with the suppression intensifying as the concentration increased [48]. Additionally, dibutyl phthalate significantly inhibited the seed germination in *Nitraria roborowskii* [49]. Our study found that dibutyl phthalate, 2,6-di-tert-butylphenol, and 2,4-di-tert-butylphenol significantly suppressed the germination of *B. rapa* seeds, with their inhibitory effects intensifying as concentrations increased. The results suggest that these three inhibitory substances are responsible for low germination rate of *C. monnieri* seeds. This may be because phenolic compounds such as 2,6-di-tert-butylphenol and its isomers can induce oxidative stress, leading to lipid peroxidation and compromising the integrity of cell membranes–a process critical for radicle emergence and seed germination [50]. Furthermore, dibutyl phthalate disrupts energy metabolism and enzymatic activity, thereby inhibiting key metabolic processes and depriving the developing embryo of essential energy and carbon skeletons in plant seeds [51].

## 5. Conclusions

Our study demonstrated that *C. monnieri* seeds maintained high viability with relatively low germination percentage, confirming that low germination rate was mainly due to MPD rather than seed vigor (even mortality). Moreover, seed-coat treatment significantly enhanced germination percentage and germination speed index (GSI) of *C. monnieri* seeds, implying that the seed coat of *C. monnieri* acts as a mechanical constraint that impedes the protrusion of the morphologically mature radicle, thereby preventing seed germination of *C. monnieri*. Furthermore, the diethyl ether and ethyl acetate phase extracts from *C. monnieri* seeds significantly inhibited seed germination of *B. rapa*, containing endogenous inhibitory substances in *C. monnieri* seeds. And the mainly endogenous inhibitory substances were dibutyl phthalate, 2,6-di-tert-butylphenol, and 2,4-di-tert-butylphenol with significant higher inhibitory effects on seed germination and seedling growth of *B. rapa*. Our study showed that seed coat constraints and endogenous inhibitory substances collectively influence the germination of *C. monnieri* seeds, providing a mechanistic basis for developing effective germination-promoting techniques. Based on this mechanistic understanding of morphophysiological dormancy (MPD) of plant seeds, the logical next step is to develop optimized stratification or chemical treatment protocols that can effectively overcome seed dormancy, thereby facilitating large-scale cultivation of *C. monnieri*.

## Figures and Tables

**Figure 1 plants-14-03801-f001:**
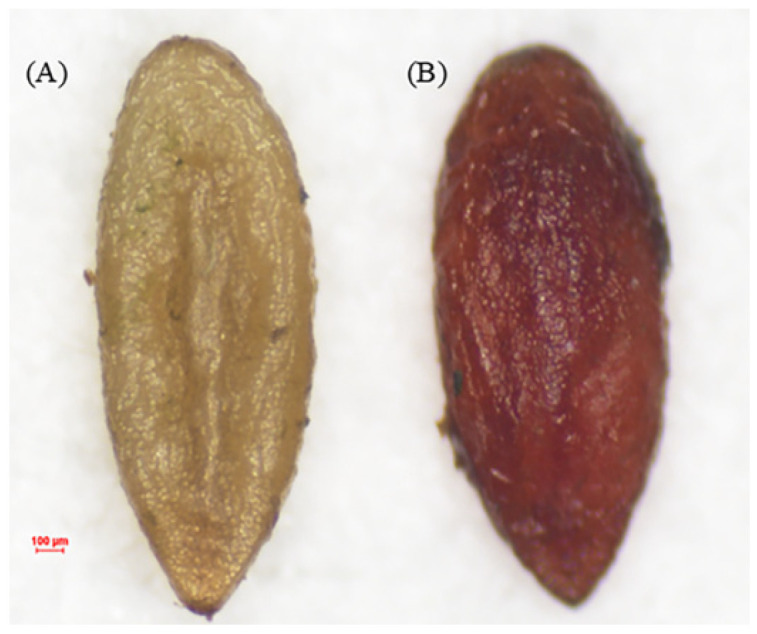
The non-viable ((**A**); uncolored) and viable ((**B**); stained red) seeds of *C. monnieri* tested by using the 2,3,5-triphenyltetrazolium chloride (TTC) method.

**Figure 2 plants-14-03801-f002:**
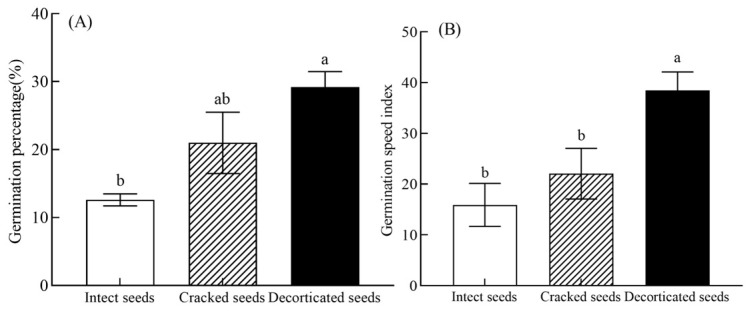
Effects of different seed-coat treatments on the germination percentage (**A**) and germination speed index (**B**) of *C. monnieri* seeds. Values are the means ±SE of 4 replicates. In each panel, means with different lowercase letters were significantly different among the seed-coat treatments by Tukey’s test at *p* < 0.05.

**Figure 3 plants-14-03801-f003:**
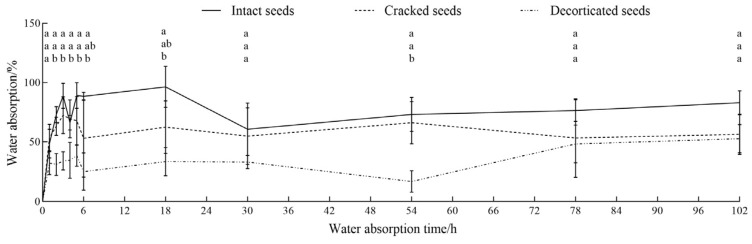
Changes in water absorption rates of the intact, cracked and decorticated seeds of *C. monnieri.* Different lowercase letters indicated significant difference among the seed-coat treatments by Tukey’s test at *p* < 0.05.

**Figure 4 plants-14-03801-f004:**
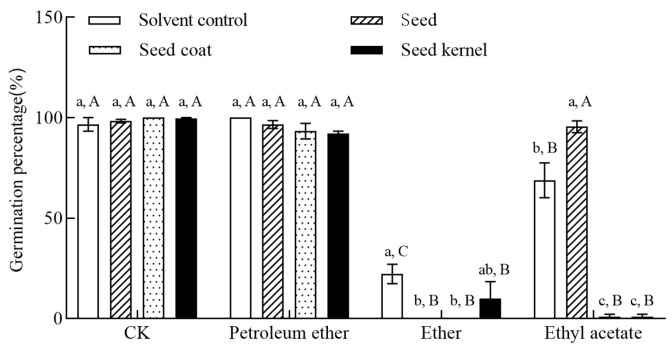
Inhibitory effects of different organic solvent extracts from different parts of *C. monnieri* seeds on the germination percentage of non-dormant *B. rapa* seeds for the germination inhibition bioassay. Different lowercase and uppercase letters indicated significant differences among different seed part treatments for the same organic phase, and among different organic phase extracts for the same seed part by Tukey’s test at *p* < 0.05, respectively. CK referred to the aqueous solution.

**Figure 5 plants-14-03801-f005:**
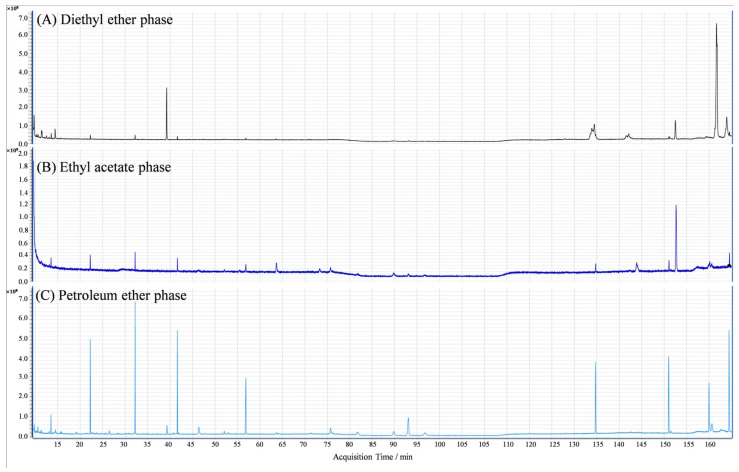
GC-MS identification chromatograms of the components in diethyl ether phase (**A**), ethyl acetate phase (**B**), and petroleum ether phase (**C**) extracted from *C. monnieri* seeds.

**Figure 6 plants-14-03801-f006:**
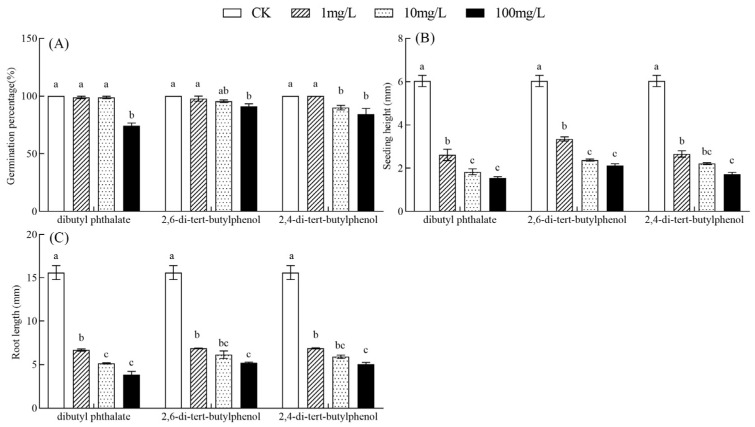
Effects of three inhibitory substances at different concentrations on the germination index of *B. rapa* seeds. Germination percentage of *B. rapa* seeds (**A**); Seedling heights of *B. rapa* (**B**); Root length of *B. rapa* (**C**). In each panel, means with different lowercase letters were significantly different among the treatments with different concentrations of the same inhibitory substance by Tukey’s test at *p* < 0.05.

**Table 1 plants-14-03801-t001:** Results of the ANOVAs for the effects of seed-coat treatment (cracked, decorticated, and intact seeds) on the germination percentage, germination speed index (GSI) and water absorption rate of *C. monnieri* seeds.

Measured Indexes	Water Absorption Time	F	*p*
Germination percentage (%)	/	7.86	0.007 **
Germination speed index	/	7.25	0.009 **
Water absorption rate (%)	1 h	3.14	0.12
	2 h	22.85	0.002 **
	3 h	17.41	0.003 **
	4 h	6.44	0.032 *
	5 h	9.64	0.013 *
	6 h	6.91	0.028 *
	18 h	9.63	0.013 *
	30 h	1.76	0.25
	54 h	14.06	0.005 *
	78 h	1.42	0.31
	102 h	4.65	0.060

Note: “/”: no data; * *p* < 0.05; ** *p* < 0.01.

**Table 2 plants-14-03801-t002:** Two-way ANOVAs for the effects of different organic phase extracts and different seed parts of *C. monnieri* on the germination percentage (%) of *B. rapa*.

Measured Indexes	Seed Part		Phase Type		Seed Part × Phase Type	
	F	*p*	F	*p*	F	*p*
Germination percentage (%)	47.398	<0.001	548.084	<0.001	42.278	<0.001

**Table 3 plants-14-03801-t003:** Types and relative contents of the organic phase extracts from *C. monnieri* seeds.

Extracted Substances	Molecular Formula	Type of Organic Phases					
		Diethyl Ether Phase		Ethyl Acetate Phase		Petroleum Ether Phase	
		Retention Time (min)	Relative Percent (%)	Retention Time (min)	Relative Percent (%)	Retention Time (min)	Relative Percent (%)
2,6,10-trimethyl-tetradecane	C_17_H_36_	/	/	/	/	14.47	2.25
6-methyl-hexadecane	C_19_H_40_	/	/	/	/	15.63	1.53
Methoxyacetic acid,2-tridecyl ester	C_16_H_32_O_3_	/	/	/	/	20.91	2.28
Isobornyl acetate	C_12_H_20_O_2_	/	/	/	/	26.48	4.20
2,6-di-tert-butyl-phenol	C_14_H_22_O	39.28	39.78 a	/	/	39.31	10.43
Perilla alcohol	C_10_H_16_O	/	/	/	/	46.39	18.51
9,9′-[1,4-phenylenedi-2,1-ethenediyl]bis[2,3,6,7-tetrahydro-1H,5H-benzo[ij]quinolizine	C_34_H_36_N_2_	/	/	53.78	8.27 b	58.36	2.44
Methyl oleate	C_19_H_36_O_2_	/	/	/		71.71	9.06
1,2-15,16-diepoxyhexadecane	C_16_H_30_O_2_	/	/	89.81	3.54 b	81.73	19.73
Cis-p-mentha-1(7),8-dien-2-ol	C_10_H_16_O	/	/	/	/	96.71	10.50
Osthole	C_15_H_16_O_3_	/	/	160.37	11.60 b	160.61	19.08
2,4-di-tert-butylphenol	C_14_H_22_O	/	/	63.66	7.44 b	/	/
1,3-diacetoxy-2-propanyl laurate	C_19_H_34_O_6_	/	/	151.34	2.18 b	/	/
Dibutyl phthalate	C_16_H_22_O_4_	152.64	22.08 ab	152.63	47.03 a	/	/
p-xylene	C_8_H_10_	11.37	13.76 ab	/	/	/	/
Cyclooctatetraene	C_8_H_8_	14.40	3.63 b	/	/	/	/
Methyl 14-(2-octylcyclopropyl) tetradecanoate	C_26_H_50_O_2_	151.03	2.20 b	/	/	/	/
Cis-octacecane-9-enoic acid	C_41_H_64_O_13_	164.48	1.10 b	/	/	/	/
One-way ANOVA (*F*/*p* values)			2.47/0.092		2.70/0.069		1.12/0.39

Note: “/” indicated that the compound was not detected. Different lowercase letters indicated significant differences between the substances in the same organic phase extract by Tukey’s test at *p* < 0.05.

## Data Availability

The data obtained in this study are presented “as is” in at least one of the figures or tables embedded in the manuscript.

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
