# Peer review of "Analysis of the Factors Affecting Germination of Cnidium monnieri Seeds and Its Endogenous Inhibitory Substances"

_plants, 2025, doi:10.3390/plants14243801_

Round 1

Reviewer 1 Report

Comments and Suggestions for Authors

The manuscript is very well structured and presents clear and relevant results. However, I suggest including additional practical germination analyses to strengthen the applied value of the study and expanding the physiological interpretation of the results to better explain the mechanisms involved in seed dormancy. Overall, the work is of high quality and makes an important contribution to the topic.

Author Response

Comment: The manuscript is very well structured and presents clear and relevant results. However, I suggest including additional practical germination analyses to strengthen the applied value of the study and expanding the physiological interpretation of the results to better explain the mechanisms involved in seed dormancy. Overall, the work is of high quality and makes an important contribution to the topic.

Response: Thanks for your comment! We fully agree with these suggestions and have revised the manuscript accordingly to address them.

Regarding "including additional practical germination analyses": In response to this suggestion, we have incorporated the Germination Speed Index (GSI) as a key metric alongside the final germination percentage. This analysis provides a more dynamic and practical assessment of germination, which is crucial for evaluating seed vigor and predicting field emergence uniformity. We revised this paragraph (L162 - L168: L254 - L257; L372 - L378).

Regarding "expanding the physiological interpretation of the results": We have substantially expanded the physiological interpretation in the Discussion section to provide a deeper mechanistic explanation of our findings.

Specifically: Firstly, we provide a more detailed discussion on how the mechanical constraint of the seed coat physically impedes embryo expansion and radicle protrusion, interpreting the significant increase in both germination percentage and GSI after coat removal as direct evidence of this mechanism. We offer a physiological explanation for the water absorption results, arguing that the intact seed coat's role is primarily mechanical rather than a barrier to water uptake.

Secondly, we elaborate on the potential physiological modes of action of the identified key inhibitory substances (dibutyl phthalate and 2,6-di-tert-butylphenol), suggesting they may induce oxidative stress and disrupt energy metabolism, thereby suppressing germination.

We believe that these revisions have thoroughly addressed the Reviewer's valuable suggestions, enhancing both the practical relevance and the scientific depth of our manuscript. We revised this paragraph (L380 - L385; L397 - L400; L401 - L409; L467 - L474).

Reviewer 2 Report

Comments and Suggestions for Authors

Please consider replacing the word “split”. This term is usually employed to express something that was separated in two halves. In this case, the reader can be misled to think that the seed was separated in two parts. Maybe the word “broken” would express it better or you may find a better word.

Although this is not a key result, it is difficult to understand why the water absorption was lower in the split and decorticated seeds in comparison to the control treatment. The quick explanation of damages made to the structure is not convincing (line 360 in the Discussion Section). Some damage to the seed would not change its water absorption profile because that is just a physical process (it is not a physiological process). Was any part of the seed removed in those two treatments? That could be a reason to alter the water absorption rate because it change the ratio of seed weight to water. This was the only possibility that I could figure out.

Comments on the Quality of English Language

Line 89: Please confirm if “sprouting” is the correct term. Usually sprouting refers to buds, and this is referring to seeds.

Line 91: “…separation of different inhibitory substances…”

Line 147: “…seeds were positioned with the radicle oriented…”

Line 182: (typo) inhibitory

Figure 4: please check the uppercase letter over the bar in the solvent control of Ether extract.

Line 369: “…studies of some inhibitory…”

Author Response

Comment 1: 

Q1: Please consider replacing the word “split”. This term is usually employed to express something that was separated in two halves. In this case, the reader can be misled to think that the seed was separated in two parts. Maybe the word “broken” would express it better or you may find a better word.

Response: Thanks! we fully agree with the reviewer's point that the term "split" could be ambiguous and might misleadingly suggest the seed was divided into symmetrical halves. As suggested, we have replaced "split" with the term "broken" throughout the manuscript.

Q2: Although this is not a key result, it is difficult to understand why the water absorption was lower in the split and decorticated seeds in comparison to the control treatment. The quick explanation of damages made to the structure is not convincing (line 360 in the Discussion Section). Some damage to the seed would not change its water absorption profile because that is just a physical process (it is not a physiological process). Was any part of the seed removed in those two treatments? That could be a reason to alter the water absorption rate because it change the ratio of seed weight to water. This was the only possibility that I could figure out.

Response: We are grateful to the reviewer for their insightful comment, which is absolutely correct. As the reviewer rightly pointed out, water absorption is inherently a physical process. However, its rate and extent are highly dependent on the compositional properties of the seed. The different components of a seed are not homogeneous; they possess distinct physical structures and chemical compositions, leading to significant differences in their water absorption capacity per unit mass. The removed portion (seed coat) is typically rich in hydrophilic dietary fibers such as cellulose and hemicellulose, with a porous structure that serves as the primary pathway and initial reservoir for rapid water entry and storage. The remaining portion (embryo and inner endosperm), often rich in lipids (hydrophobic) or encapsulated within denser starch/protein matrices, generally exhibits an intrinsic water absorption capacity and rate lower than that of the seed coat.

Therefore, we fully agree that our original explanation in the manuscript, which attributed the phenomenon merely to "structural damage," was vague and unconvincing. The reviewer's comments prompted us to reconsider this more deeply.

In the revised manuscript, we have rewritten the relevant discussion section, shifting the focus from the general "structural damage" to the more specific explanation: "The removal of the highly water-absorbent seed coat tissue during the decortication process resulted in a decrease in the water absorption capacity per unit dry weight of the remaining seed tissue, thereby leading to the observed reduction in the overall water absorption rate." We revised this paragraph (L397 - L409).

Comment 2:

Q1: Line 89: Please confirm if “sprouting” is the correct term. Usually sprouting refers to buds, and this is referring to seeds.

Q2: Line 91: “…separation of different inhibitory substances…”

Q3: Line 147: “…seeds were positioned with the radicle oriented…”

Q4: Line 182: (typo) inhibitory

Q5: Figure 4: please check the uppercase letter over the bar in the solvent control of Ether extract.

Q6: Line 369: “…studies of some inhibitory…”

Response: We thank the reviewer for this precise correction. 

About Q1, the reviewer is absolutely right that "sprouting" is more appropriately used to describe the emergence of buds or shoots from vegetative tissues, whereas "germination" is the correct term for the process by which a seed develops into a new plant. We have therefore replaced "sprouting inhibitors" with the accurate term "germination inhibitors" to ensure terminological precision.

About Q2, following the reviewer's suggestion, we have revised the sentence to read: "The extraction process typically involves leaching, followed by systematic extraction through the separation of different inhibitory substances into distinct solvent phases based on their solubility." We believe this revision more directly and clearly describes the process.

About Q3, we have revised the sentence as suggested to ensure methodological clarity. The sentence now reads: "Simultaneously, all seeds were positioned with the radicle oriented downward on Petri dishes lined with three layers of filter paper. "

About Q4, the word "inhibitory" has been correctly spelled in the revised manuscript. We thank the reviewer for their careful attention to detail.

About Q5, we have checked and corrected the lettering over the bars in Figure 4.

About Q6, we have revised this sentence to incorporate the suggested phrase and to improve the overall flow. The sentence in the discussion section now reads: "On the other hand, studies of some inhibitory substances in seeds have shown them to be a significant factor that impedes normal germination"

Reviewer 3 Report

Comments and Suggestions for Authors

All suggestions for improving the manuscript have been noted within the text. The paper is interesting and provides valuable information. However, it lacks details regarding the seed germination process, specifically the duration and conditions of stratification that would result in the highest germination rate. Including this information would significantly enhance the scientific value and practical applicability of the study.

Author Response

Comment 1: All suggestions for improving the manuscript have been noted within the text. The paper is interesting and provides valuable information. However, it lacks details regarding the seed germination process, specifically the duration and conditions of stratification that would result in the highest germination rate. Including this information would significantly enhance the scientific value and practical applicability of the study.

Response: We thank the reviewer for their positive assessment of our work and for raising this important point regarding stratification. We agree that identifying the optimal pre-treatment to break dormancy is key for practical application.

In this study, our primary objective was to identify the fundamental causes of seed dormancy in C. monnieri (i.e., the mechanical barrier and endogenous inhibitors), rather than to optimize a specific germination protocol. The experiments were therefore designed to elucidate the underlying mechanisms under controlled, standardized conditions. However, in direct response to the reviewer's comment, we have taken the following actions to address this valuable suggestion:

We have added a sentence in Section 2.2.2 to explicitly state the germination conditions used in our assay: "For the germination assays, seeds were not stratified in order to evaluate their innate dormancy under standard optimal conditions." This clarifies that our reported germination percentages (12.6% for intact seeds) represent the baseline dormancy level without pre-treatment.

Enhanced Discussion and Future Perspective: We have expanded the Discussion and Conclusion sections to directly acknowledge this point and outline the logical next steps. We revised this paragraph (L154 - 156; L513 - L516).

Reviewer 4 Report

Comments and Suggestions for Authors

The article presents a study on seed dormancy in Cnidium monnieri. The authors discuss the type of dormancy and the substances that inhibit germination. The article is relevant because the species is important to its region. Some questions and suggestions can be found in the comments section of the text. I suggest that the authors correct the suggested points to improve the article.  

Author Response

Comment : The article presents a study on seed dormancy in Cnidium monnieri. The authors discuss the type of dormancy and the substances that inhibit germination. The article is relevant because the species is important to its region. Some questions and suggestions can be found in the comments section of the text. I suggest that the authors correct the suggested points to improve the article.

Q1: If possible, calculate the germination speed index (Maguire, 1973); that would be valuable data for the study.

Q2: Limited physiological interpretation. improve

Q3: put in the rules

Q4: rules

Response: We thank the reviewer for these insightful suggestions, which have helped us significantly enhance the depth and quality of our manuscript. We have addressed all comments as detailed below.

About Q1, we agree with the reviewer that this is a valuable addition. Following the suggestion, we have calculated the Germination Speed Index (GSI) according to Maguire's formula. We revised this paragraph (L160 - L168: L254 - L257; L372 - L378).

About Q2, we sincerely thank the reviewer for this critical insight. Accordingly, we have thoroughly revised the Discussion section to provide a more mechanistic interpretation of our findings. The key improvements include: We now elaborate on the specific physiological consequences of this constraint. We explicitly state that the rigid seed coat physically restricts embryo expansion and radicle emergence, thereby preventing germination even when the embryo is fully developed and imbibed. The significant increase in both final germination percentage and GSI following coat removal is presented as direct evidence that the primary role of the coat is to provide this mechanical resistance, not to inhibit water uptake. And we propose specific physiological and biochemical modes of action for the key inhibitors identified (dibutyl phthalate and 2,6-di-tert-butylphenol). We revised this paragraph (L380 - L385; L397 - L409; L467 - L474).

About Q3 and Q4, We thank the reviewer for this comment. We have carefully checked and revised the reference list to ensure that all citations now conform to the required journal format. The necessary changes have been made throughout the manuscript.
